# Piceatannol Prevents Obesity and Fat Accumulation Caused by Estrogen Deficiency in Female Mice by Promoting Lipolysis

**DOI:** 10.3390/nu15061374

**Published:** 2023-03-12

**Authors:** Kotoko Arisawa, Miyuki Kaneko, Ayumi Matsuoka, Natsuki Ozawa, Rie Kawawa, Tomoko Ishikawa, Ikuyo Ichi, Yoko Fujiwara

**Affiliations:** 1Graduate School of Humanities and Sciences, Ochanomizu University, Tokyo 112-8610, Japan; kotoko.arisawa.b3@tohoku.ac.jp (K.A.); g2140539@edu.cc.ocha.ac.jp (A.M.); g2240523@edu.cc.ocha.ac.jp (N.O.); ichi.ikuyo@ocha.ac.jp (I.I.); 2Graduate School of Pharmaceutical Sciences, Tohoku University, Sendai 980-8577, Japan; 3Institute for Human Life Science, Ochanomizu University, Tokyo 112-8610, Japan; ishikawa.tomoko@wa.seitoku.ac.jp; 4Department of Human Nutrition, Seitoku University, Chiba 271-8555, Japan

**Keywords:** piceatannol, obesity, estrogen deficiency, menopause

## Abstract

Postmenopausal women have a higher susceptibility to obesity and chronic disease. Piceatannol (PIC), a natural analog of resveratrol, was reported to inhibit adipogenesis and to have an antiobesity effect. In this study, PIC’s effect on postmenopausal obesity and the mechanism of its action were investigated. C57BL/6J female mice were divided into four groups and half of them were ovariectomized (OVX). Both OVX and sham-operated mice were fed a high-fat diet (HFD) with and without the addition of 0.25% of PIC for 12 weeks. The abdominal visceral fat volume was higher in the OVX mice than the sham-operated mice, and PIC significantly decreased the fat volume only in the OVX mice. Unexpectedly, expression levels of adipogenesis-related proteins in white adipose tissue (WAT) were suppressed in the OVX mice, and PIC did not affect lipogenesis in either the OVX or sham-operated mice. Regarding the expression of proteins associated with lipolysis, PIC activated the phosphorylation of hormone-sensitive lipase much more in the OVX mice, but it did not affect the expression of adipose triglyceride lipase. PIC also tended to induce the expression of uncoupled protein 1 in brown adipose tissue (BAT). These results suggest that by promoting lipolysis in WAT and deconjugation in BAT, PIC is a potential agent to inhibit fat accumulation caused by menopause.

## 1. Introduction

Postmenopausal women are prone to obesity resulting from estrogen deficiency [1,2]. Menopausal changes tend to occur in fat deposition rather than in body weight gain and cause visceral fat accumulation, whereas premenopausal women accrue subcutaneous fat [3,4]. Estrogen deficiency is known to lead to many metabolic changes, such as overeating as a result of resistance to leptin sensitivity [5,6], low metabolic expenditure [3], and a decrease in spontaneous activity [6]. Excessive accumulation of visceral fat is known to increase the risks associated with metabolic syndromes, such as cardiovascular diseases, atherosclerosis, hypertension, and type 2 diabetes [7,8,9,10]. Therefore, suppressing the excessive accumulation of visceral fat is important for postmenopausal women’s health. 

Piceatannol (3,5,3′,4′-tetrahydroxystilbene, PIC) is a natural analog of resveratrol (RSV) [11], one of the widely studied polyphenols that have preventive effects on the development of obesity-related diseases [12,13,14]. PIC is found in the seeds of passion fruit, grapes, and many other fruits [15,16,17] and is known as a specific inhibitor of spleen tyrosinase kinase (Syk) [18]. Previous studies have identified many biological activities of PIC, including antioxidant [19], anticancer [20], and anti-inflammatory effects [11,21,22]. Especially, PIC is reported to inhibit adipogenesis of 3T3-L1 cells [12], mesenchymal stem cells [23], and cells from human visceral adipose tissue [24]. The inhibitory effect of PIC on adipocyte differentiation is reportedly mediated by the modulation of mitotic clonal expansion and insulin signaling in the early stage of differentiation [12,23,24]. In vivo, PIC has also been shown to reduce adipose tissue weight in male C57BL/6 mice fed a high-fat diet (HFD) and to suppress obesity complications in Zucker rats [13,14]. However, it is not clear whether PIC has a beneficial effect on obesity in postmenopausal women.

Sex-specific disparities are commonly observed in animal models of obesity, often with a greater degree of severity in males [25]. The C57BL/6J mouse strain is widely used in obesity research and has been shown to develop obesity in male mice when fed an HFD. In contrast, female C57BL/6J mice exhibit greater resilience to the obesogenic effects of an HFD [26]. However, when subjected to ovariectomy, ERα knockout, or aromatase knockout, these female mice can develop obesity when fed an HFD [27,28,29]. Hormone replacement therapy has been found to suppress obesity resulting from estrogen deficiency [30]. These findings suggest that sex hormones play a protective role against obesity in female mice, and these mouse models are being utilized to investigate the mechanisms underlying postmenopausal obesity.

Stilbene compounds, such as RSV and PIC, have similar molecular structures to 17 beta-estradiol [31], suggesting the possibility that they could act as estrogen replacement therapy. Hence, our study evaluated the effect of PIC on obesity induced by estrogen deficiency using ovariectomized (OVX) mice. Our findings indicate that PIC reduces fat accumulation in OVX mice by promoting lipolysis.

## 2. Materials and Methods

### 2.1. Materials

Piceatannol was obtained from Morinaga Research Institute (Tokyo, Japan) in 2017. Its concentration was measured using HPLC analysis as previously described [32].

### 2.2. Animals and Experimental Design

The Animal Ethics Committee of Ochanomizu University approved the study (No. 18009, 1 May 2018; and 20028, 15 October 2020). Six-week-old female C57BL/6J mice were purchased from CLEA Japan, Inc. (Tokyo, Japan) and housed in cages with a maximum of five mice per cage. The mice were maintained in a room with controlled temperature (21–25 °C) and humidity (40–60%) with a 12 h dark/light cycle under artificial lighting. During acclimation, the mice were fed a regular chow diet (CLEA Rodent Diet CE-2, CLEA Japan, Inc.) and had free access to water.

The design of the experiment is shown in Appendix A. After a 7-day acclimation period, the mice were divided into 2 groups based on their body weight and were ovariectomized (OVX) or sham-operated (sham) (*n* = 15 and 12 for the OVX and sham groups, respectively). The surgery was performed under deep anesthesia following intraperitoneal injection of a 3-mixed solution (0.1 mL/10 g B.W.) containing Domitor^®^ (0.3 mL; Nippon Zenyaku Kogyo Co, Ltd., Fukushima, Japan), midazolam (0.8 mL; Fuji Pharma Co., Ltd., Toyama, Japan), Vetorphale^®^ (1.0 mL; Meiji Seika Pharma Co, Ltd., Tokyo, Japan), and physiological saline (7.9 mL). For the OVX group, the anterior uterine horns were excised to remove the ovaries, while in the sham-operated controls, the two ovaries were visualized but not removed [27,33].

The control diet used in this study was based on the AIN-93M diet and had a high-fat content (HFD, 45% fat, 20% protein, and 35% carbohydrate). The experimental group was fed the same HFD but with the addition of 0.25% PIC. Detailed information on the composition of the experimental diets used in this study is described in Appendix A. After one week of either sham or OVX surgery, the mice were assigned randomly to one of four groups and placed on either the control or experimental diet. The sham and OVX groups were each divided into an HFD-alone group and an HFD containing 0.25% PIC group (sham, sham + PIC, OVX, and OVX + PIC). Throughout the experimental period, all mice were provided with unlimited access to water and diets. After 12 weeks, all mice were euthanized following administration of pentobarbital sodium (Somnopentyl; Kyoritsu Seiyaku Corporation, Tokyo, Japan) into the abdominal cavity. Under anesthesia, blood was collected from the axillary vein, and tissues were harvested and stored at −80 °C until further analysis. Mice that did not show uterine atrophy upon dissection after the 12-week experiment were excluded from the results. The experimental period was determined based on our unpublished previous studies and was the period during which obesity due to the HFD was reliably observed in the OVX group compared to the sham group.

### 2.3. Glucose Tolerance Test

The glucose tolerance test was performed 11 weeks after the start of the experimental diet. After 16 h fasting period, fasting blood glucose concentration was measured from the tail using FreeStyle Precision Neo/FS precision blood glucose test strips (Abbott Japan LLC, Tokyo, Japan). Subsequently, an intraperitoneal injection of a 10% glucose solution was administered at a dose of 1 mg/g of body weight. After administering glucose, blood glucose levels were evaluated at 15, 30, 60, and 120 min.

### 2.4. Biochemical Parameters of Blood

Blood was collected from the axillary vein of the mice under anesthesia at the time of dissection, mixed with a small amount of heparin sodium (5000 units/5 mL; Mochida Pharmaceutical Co., Ltd., Tokyo, Japan), and placed on ice. After centrifugation at 1200× *g* for 15 min, the resulting supernatant plasma was collected. Plasma total cholesterol and free fatty acid concentrations were determined with an enzymatic colorimetric assay, entrusted to Oriental Yeast Co., Ltd. (Tokyo, Japan). Plasma triglyceride concentrations were measured using enzymatic methods (Triglyceride E-Test Wako; FUJIFILM Wako Pure Chemical Corporation, Osaka, Japan).

### 2.5. Micro X-ray Computed Tomography (CT) Analysis of Fat Accumulation

To assess fat deposits, the abdominal region between the first and fifth lumbar vertebrae of the mouse was analyzed using micro-X-ray CT (CosmoScan FX; Rigaku, Tokyo, Japan). Images were acquired with a tube voltage of 90 kV, tube current of 88 μA, 120 μm slice thickness, and total scan time of 2 min [34,35]. Following scanning, the visceral and subcutaneous fat mass were calculated using 3-dimensional image analysis software (Analyze 12.0; AnalyzeDirect, Overland Park, KS, USA). The abdominal muscular wall was used to separate the visceral fat (inside of the abdominal wall) from the subcutaneous fat (outside of the abdominal wall) [36]. These analyses were performed at 4, 8, and 12 weeks after the start of the experimental diet.

### 2.6. Western Blotting

Proteins were extracted from the tissues using RIPA buffer (R0278; Sigma-Aldrich, St. Louis, MO, USA) supplemented with protease inhibitor cocktail (P8340; Sigma-Aldrich) and phosphatase inhibitor cocktails (P5726 and P0044; Sigma-Aldrich). Protein concentrations were determined using the BCA assay (FUJIFILM Wako Pure Chemical Corporation). Equivalent amounts of proteins were separated by electrophoresis on 8% SDS-PAGE and subsequently transferred to Immobilon PVDF membranes (Bio-Rad Laboratories, Inc., Hercules, CA, USA). The membranes were treated with blocking reagent (5% *w*/*v* nonfat dry milk in Tris-buffered saline containing 0.1% Tween-20) for 1 h at room temperature. After that, the membranes were incubated overnight at 4 °C with the following primary antibodies (all from Cell Signaling Technology, Danvers, MA, USA): rabbit antiphospho-HSL (Ser563; CST#4139), rabbit anti-HSL (CST#18381), rabbit anti-ATGL (CST#2439), rabbit anti-p-ACC (Thr 172; CST#2535), rabbit anti-ACC (CST#3662), rabbit anti-FAS (CST#3189), and mouse anti-β-actin (CST#8457). The membranes were then incubated with peroxidase-conjugated antirabbit antibody (Jackson ImmunoResearch Laboratories, West Grove, PA, USA; 111-035-144) or antimouse secondary antibody (Jackson ImmunoResearch Laboratories; 115-005-003). Immunoreactivity was detected using ECL Prime western blotting detection reagents (Cytiva, Tokyo, Japan) with β-Actin serving as the loading control.

### 2.7. Quantitative Real-Time PCR (qPCR)

Total RNA was extracted from frozen tissues using TRIzol reagent (ThermoFisher Scientific, Cleveland, OH, USA) and reverse transcribed into cDNA with ReverTra Ace^®^ qPCR RT Master Mix (TOYOBO CO., LTD., Osaka, Japan). Gene expression levels were assessed by real-time PCR on the StepOnePlus Real-Time PCR System (Thermo Fisher Scientific) with SYBR Green (THUNDERBIRD^®^ SYBR qPCR Mix, TOYOBO). β-actin mRNA served as the invariant control. The primer sequences used are as follows: β-actin (forward, 5′-ACT-ATT-GGC-AAC-GAG-CGG-TT-3′; reverse, 5′-ATG-GAT-GCC-ACA-GGA-TTC-CA-3′), UCP1 (forward, 5′-GGA-TGG-TGA-ACC-CGA-CAA-CT-3′; reverse, 5′-GAT-CTG-AAG-GCG-GAC-TTT-GG-3′), PGC1α (forward, 5′-AAG-TGT-GGA-ACT-CTC-TGG-AAC-TG-3′; reverse, 5′-GGG-TTA-TCT-TGG-TTG-GCT-TTA-TG-3′) and SIRT1 (forward, 5′-ATG-ACG-CTG-TGG-CAG-ATT-GTT-3′; reverse, 5′-CCG-CAA-GGC-GAG-CAT-AGA-T-3′). The thermal cycling conditions employed in this study consisted of an initial denaturation step at 95 °C for 10 min followed by 40 cycles at 95 °C for 15 s and 60 °C for 1 min.

### 2.8. Statistical Analysis

All data were reported as the mean ± standard error (M ± SE). Statistical analysis was performed using Microsoft Excel for Mac version 16.70 and one-way ANOVA with GraphPad Prism 8 software (GraphPad Software Inc., San Diego, CA, USA) followed by the Tukey–Kramer’s test. A *p*-value below 0.05 was considered statistically significant.

## 3. Results

### 3.1. Effect of Piceatannol on Body and Tissue Weight in HFD-Fed Ovariectomized C57BL/6J Mice

In this study, acclimated seven-week-old C57BL/6J mice were subjected to OVX or sham surgery. Following 1 week of acclimation, the mice were divided into 2 groups randomly to ensure that the sham and OVX groups had the same baseline body weight, after which they were fed a 45% high-fat diet (HFD) or the same HFD but with the addition of 0.25% piceatannol (PIC). The sham-HFD, sham-PIC, OVX-HFD, and OVX-PIC groups contained five, six, six, and six mice, respectively.

Figure 1a shows the results of the weekly weight measurements. At the end of the experiment, the OVX-HFD group was approximately 30% heavier than the sham-HFD group, indicating that OVX increases obesity, as seen in previous reports [27]. On the other hand, the OVX-PIC group had a significant reduction in weight compared to the OVX group. However, there was no notable difference between the sham + PIC and sham groups. The results indicated that PIC had a selective effect on OVX-induced obesity and did not reduce the body weight of female mice who were not obese. Food intake among the four groups did not differ significantly, as shown in Figure 1b. White adipose tissue (WAT), brown adipose tissue (BAT), and liver weights are shown in Figure 1c. OVX significantly increased WAT weight, and PIC suppressed the increase; in BAT and the liver, there was no increase in weight by OVX. However, in BAT, the PIC-supplemented groups among both the sham and OVX mice showed a decreased weight. Browning with fat accumulation in the tissue was observed in the BAT of sham-HFD and OVX-HFD. These findings propose that PIC supplementation reduces OVX-induced weight gain by decreasing the weight of adipose tissue.

### 3.2. Effects of PIC on Blood Biochemical Parameters in Ovariectomized Mice

The OVX + HFD group had significantly elevated serum total cholesterol levels compared to the sham + HFD group, but PIC supplementation significantly suppressed this effect (Figure 2a). Plasma triglyceride and nonesterified fatty acid levels in any group differed significantly from one another (Figure 2b,c). None of the groups displayed a substantially elevated blood glucose level during the oral glucose tolerance test, and the administration of PIC had no influence on blood glucose or insulin levels (Figure 2d,e). The PIC-supplemented groups had lower areas under the blood glucose curve (AUC) compared with the HFD groups, although this distinction was not statistically significant (Figure 2f,g).

### 3.3. Effects of PIC on Visceral and Subcutaneous Fat Accumulation HFD-Fed Ovariectomized Mice

Micro-CT is noninvasive and provides high-resolution, 3D imaging of adipose tissue, enabling continuous analysis throughout a study period. The visceral and subcutaneous fat volumes of each group of mice at 4, 8, and 12 weeks, as analyzed by micro-CT, are shown in Figure 3a and Figure 3b, respectively. Fat accumulation in the OVX group was seen as early as 4 weeks and continued until 12 weeks. In each period, PIC suppressed the OVX-induced increase in fat accumulation. On the other hand, there was no significant difference between the sham + HFD and sham + PIC groups. These results suggest that PIC contributes to the suppression of obesity by inhibiting both visceral and subcutaneous fat accumulation, which was increased in OVX mice.

### 3.4. Effects of PIC on the Expression of Proteins Involved in Lipid Metabolism

The expression of proteins implicated in the metabolic processes of lipid synthesis and lipolysis in ovarian adipose tissue was evaluated by western blotting (Figure 4a,b). Even though the OVX-HFD and OVX-PIC groups showed significantly higher fat accumulation than the sham groups, protein expression levels of fatty acid synthase (FAS) and acetyl CoA carboxylase (ACC), which are associated with fat synthesis, were significantly reduced. Activated ACC catalyzes the production of malonyl-CoA; ACC is inactivated by phosphorylation. PIC did not promote ACC phosphorylation, suggesting that it does not inhibit fat synthesis.

We also assessed the evaluated levels of hormone-sensitive lipase (HSL) and adipose triglyceride lipase (ATGL), both of which are essential for the hydrolysis of triglyceride. The results showed that OVX surgery and PIC treatment did not affect the expression levels of the two lipases. However, HSL phosphorylation was decreased in the OVX-HFD group compared to the sham-HFD group, and PIC supplementation significantly improved that reduction. Phosphorylation of the serine residue (Ser563) of HSL enhances its activity and promotes lipolysis [37]. In the BAT, phospho-HSL was not increased by PIC treatment (Figure 4c). However, in terms of gene expression related to heat production, the addition of PIC had a tendency to increase UCP1 expression in the OVX group (Figure 4d). UCP1 enhances energy expenditure by promoting deconjugation in the mitochondrial inner membrane [38]. Gene expression levels of SIRT1 and PGC1α, which are regulators of UCP1, tended to increase with the addition of PIC, although no significant differences were observed. Thus, our results suggest that PIC promotes deconjugation in the BAT of postmenopausal obesity models via the SIRT1/PGC1α/UCP1 pathway [39,40]. These results suggest that PIC has antiobesity effects in OVX mice by enhancing lipolysis in WAT and energy expenditure in BAT.

## 4. Discussion

One of the important public health concerns is the increase in overweightness and obesity in menopausal women. It is known that the decreased level of estrogen is the major cause of increased obesity among menopausal women [3]. During menopause, women experience a rapid decline in ovarian function, which leads to a decrease in estrogen production. This decrease in estrogen is associated with increased overall body fat, particularly visceral fat [4]. Moreover, estrogen deficiency is reported to aggravate metabolic dysfunction and cause susceptibility to metabolic syndrome, type 2 diabetes, and cardiovascular diseases [41,42,43].

PIC, a natural polyphenolic stilbene, has been reported to exhibit various beneficial effects, such as anti-inflammatory, antioxidant, and antiproliferative activities [11]. Recent studies suggest that PIC has both in vitro and in vivo antiobesity effects. In the 3T3-L1 cell model, PIC has been shown to reduce triglyceride accumulation by inhibiting the transcription factors C/EBPβ, C/EBPα, and PPARγ [12]. In vivo studies have shown that in male C57BL/6 mice induced to become obese by an HFD, PIC inhibited weight gain; reduced blood glucose, total cholesterol, LDL, and triglyceride concentrations; and inhibited visceral fat accumulation [14]. In women, however, it was not clear whether PIC is effective for treating obesity resulting from menopause. Therefore, we examined the antiobesity effects of PIC on estrogen deficiency-induced obesity. In this study, ovariectomized (OVX) female mice were used as an estrogen-deficient obesity model that mimics menopausal women [44]. The results showed weight gain and fat accumulation in the OVX group compared to the sham group. Notably, PIC significantly suppressed OVX-induced weight gain and reduced WAT weight.

The distribution of adipose tissues differs between males and females. Males have a higher tendency to accrue fat in the visceral depot, while premenopausal females accumulate more subcutaneous fat. The tendency to deposit subcutaneous fat rather than visceral fat protects females from the harmful consequences of obesity and metabolic syndrome [5]. After menopause, the level of circulating estrogen decreases, causing a shift in the pattern of fat distribution from subcutaneous to visceral, which leads to an increase in the risk of metabolic disorders similar to that of males [4]. In the present study, PIC significantly suppressed both subcutaneous and visceral fat in the OVX group, suggesting that PIC is effective in suppressing visceral fat caused by the estrogen deficiency seen in postmenopausal women. Furthermore, micro-CT scan observations revealed that PIC had a significant antiobesity effect on OVX-induced obesity from the fourth week of the experiment, suggesting that it may also suppress early fat accumulation.

In a previous examination of PIC’s inhibitory effect on HFD-induced obesity in male C57BL/6 mice, PIC showed an antiobesity effect with higher levels of pACC and lower levels of FAS suppressing fat synthesis [14]. In the estrogen deficiency-induced obese mice used in this study, OVX surgery rather reduced fat synthesis by pACC/FAS, and PIC did not suppress the synthesis, whereas it did increase phosphorylation of HSL, the rate-limiting enzyme for lipolysis. Taken together, our results suggest that PIC suppressed fat accumulation under estrogen depletion in WAT by a completely different mechanism than that shown in previous studies. HSL is the enzyme that hydrolyzes intracellular triglyceride and diacylglycerol to fatty acids and glycerol and is activated by phosphorylation of a serine residue (Ser563), which is transferred from the cytosol to the surface of lipid droplets [45,46]. In general, when noradrenaline binds to β-adrenergic receptors (βAR), intracellular signaling pathways, such as adenylate cyclase-cAMP-protein kinase A, are activated and downstream hormone-sensitive lipases are activated to degrade neutral lipids [47]. However, it has not been reported that PIC binds and activates βARs. On the other hand, it has been reported that the cAMP/PKA pathway is one of the signaling pathways downstream of the G protein-coupled estrogen receptor (GPER) [48]. Although GPER has an estrogen-responsive sequence, it differs from the nuclear estrogen receptors ERα and ERβ in that it exerts effects through the production of second messengers rather than through transcriptional regulation by nuclear translocation [49]. Previous studies have proposed that estrogen’s role in reducing obesity may be linked to the activation of GPER. It has been reported that PIC, due to its structural similarity to diethylstilbestrol, a synthetic estrogenic agent, can act as a phytoestrogen compound [31]. The estrogen-like structure of PIC may have ameliorated estrogen-deficient obesity by increasing phospho-HSL through GPER-mediated signaling. The addition of PIC did not decrease ATGL protein, another lipase that hydrolyzes triglyceride. Although PIC reportedly reduces ATGL expression in adipocytes via upregulation of the autophagy-lysosome pathway [50], it is possible that PIC would not have affected this pathway in WAT under the conditions of this study.

Obesity is caused by excessive TG accumulation in adipocytes as well as by decreased energy expenditure and increased food intake, but in this study, there was no difference in food intake among the four groups. The decreased expression of energy expenditure-related genes in adipose tissue has been proposed as a potential factor in obesity induced by ovariectomy [27]. Mitochondrial deconjugation is a factor that increases adipocyte energy expenditure, and UCP1 promotes mitochondrial inner membrane deconjugation [38]. In this experiment, the addition of PIC tended to increase the gene expression of UCP1 in BAT. This suggests that PIC promotes deconjugation by increasing the expression of UCP1. Gene expression levels of UCP1 are regulated by SIRT1 and PGC1α, and UCP1 transcription is promoted when PGC1α is in an active state by deacetylation by SIRT1. In the present study, SIRT1/PGC1α showed a tendency to increase gene expression by PIC. In a previous report, increased gene expression of UCP1 and decreased levels of acetylation of PGC1α in BAT were observed when RSV, a PIC analog, was administered to obese mice on an HFD [39]. It has also been reported that the addition of PIC to human monocyte-derived cells increases both gene and protein expression levels of SIRT1 [51]. Figure 2a shows that PIC made no difference in nonesterified fatty acids in the blood, but it is possible that the fatty acids degraded in WAT were supplied to UCP1 in BAT, increasing energy expenditure. PGC-1α is a transcriptional coactivator that also regulates cellular energy production by promoting mitochondrial biogenesis and oxidative phosphorylation gene expression [52]. In humans, decreased PGC-1α expression is associated with obesity through reduced expression of oxidative phosphorylation genes and decreased muscle mitochondrial activity [53]. Although the increase in PGC-1α gene expression by PIC in BAT was not statistically significant in this study, future studies should further evaluate the involvement of PGC-1α-mediated enhancement of energy production in the antiobesity effects of PIC.

It has also been reported that PIC has anti-inflammatory effects: In cocultures of 3T3-L1 and RAW cells, PIC prevented fat accumulation in 3T3-L1 adipocytes by suppressing IL-6 and TNFα produced by RAW macrophage cells [21]. Thus, the possibility that PIC’s anti-inflammatory effects may also be involved in the suppression of obesity should be investigated in future studies.

Estrogen replacement therapy (ERT) is one of the treatments for postmenopausal obesity. It is known that estradiol replacement in postmenopausal model OVX rats suppresses obesity [54,55]. Estrogen replacement has also been shown to suppress obesity during menopause in human clinical studies [56]. However, the use of ERT needs to be limited because its use increases the risk of developing breast and uterine cancers; thus, there is a need to explore new treatment options [57]. The present results suggest that PIC is a useful compound for the treatment of postmenopausal obesity caused by estrogen deficiency.

## 5. Conclusions

Our findings indicate that PIC prevents OVX-induced obesity in female C57BL/6J mice. The effects of PIC are due at least in part to lipolysis associated with the activation of phospho-HSL in WAT and the modulation of the SIRT1/PGC1α/UCP1 pathway in BAT. The present findings are an important step in inhibiting fat accumulation under the state of estrogen deficiency in menopausal women. 

## Figures and Tables

**Figure 1 nutrients-15-01374-f001:**
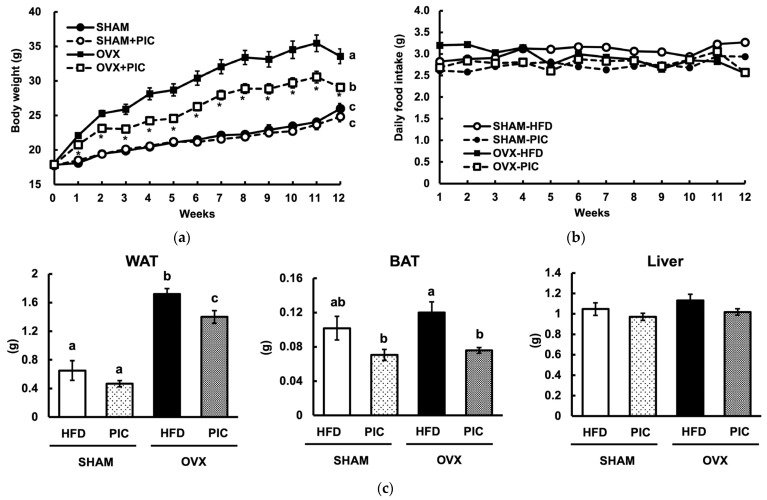
Effects of piceatannol on body weight and tissue weight in ovariectomized mice fed an HFD. (**a**) Weekly body weight measurements over 12 weeks; (**b**) daily food intake; and (**c**) weight of white adipose tissue present in the uterine cervix, interscapular brown adipose tissue, and liver. The data are presented as the means ± SE (*n* = 5–6 per group), and values with different letters are considered significantly different (*p* < 0.05) based on Tukey–Kramer’s test. Asterisk denotes significant different between the OVX-HFD and OVX-PIC groups (* *p* < 0.05).

**Figure 2 nutrients-15-01374-f002:**
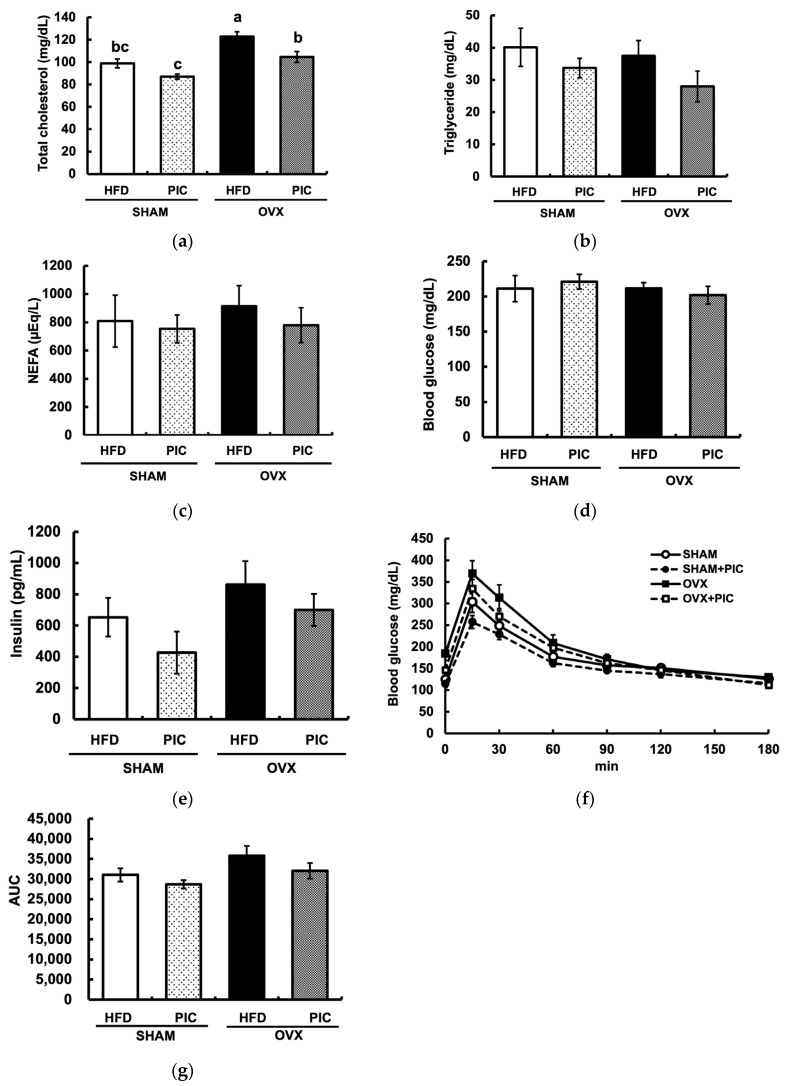
Effects of piceatannol on plasma biochemical parameters in ovariectomized mice fed an HFD. (**a**–**c**) Assessment of the serum lipid profile; (**d**) blood glucose levels and (**e**) insulin levels; (**f**) glucose tolerance test (GTT). GTT was performed one week before dissecting the mice. (**g**) area under the curve (AUC) of GTT. The data are presented as the means ± SE (*n* = 5–6 per group), and values with different letters are considered significantly different (*p* < 0.05) based on Tukey–Kramer’s test.

**Figure 3 nutrients-15-01374-f003:**
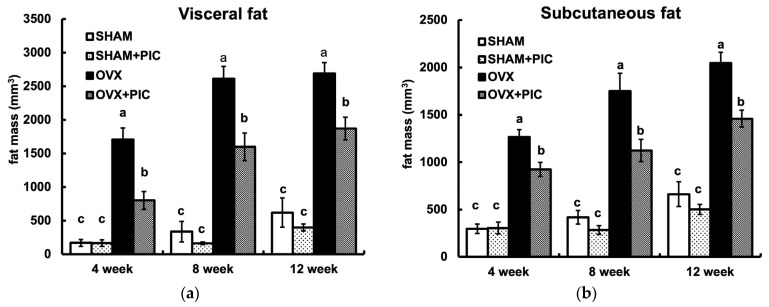
Effects of piceatannol on visceral and subcutaneous fat accumulation. (**a**) Visceral fat mass and (**b**) subcutaneous fat mass. Abdominal fat mass was measured by micro-CT scan analysis every 4 weeks. Visceral fat and subcutaneous fat were evaluated in abdominal regions between the first and fifth lumbar vertebrae. The data are presented as the means ± SE (*n* = 5–6 per group), and values with different letters are considered significantly different (*p* < 0.05) based on Tukey–Kramer’s test.

**Figure 4 nutrients-15-01374-f004:**
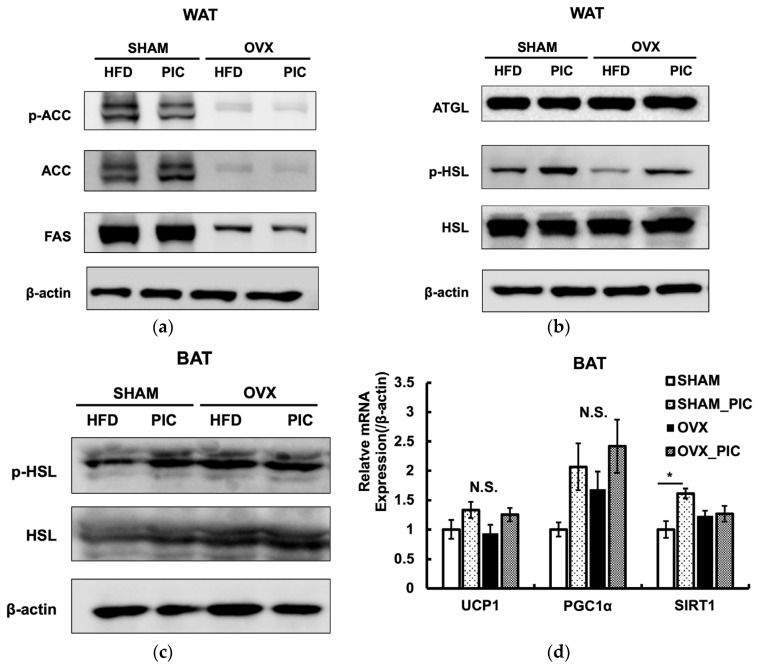
Effects of piceatannol on the expression of proteins and mRNA. (**a**) Adipogenesis-related proteins and (**b**) lipolytic proteins in WAT. (**c**) Phospho-HSL in BAT. The western blotting is representative of at least four independent experiments. (**d**) mRNA expression in BAT. Data are expressed as the means ± SE (*n* = 5–6 per group). Asterisk denotes significant difference (*p* < 0.05) as determined by Tukey–Kramer’s test and N.S. denotes not significant. The experiments were repeated three times, and the results were found to be reproducible.

## Data Availability

Data are available upon request to the corresponding author.

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
