# Peer review of "Piceatannol Prevents Obesity and Fat Accumulation Caused by Estrogen Deficiency in Female Mice by Promoting Lipolysis"

_nutrients, 2023, doi:10.3390/nu15061374_

Round 1
Reviewer 1 Report
The study is good and well writing. However, following points should be revised.
1. There are two 2.4 section, please revised it.
2. No data statistics in the manuscript. How many reduplicate assays you did?
3. I suggest the authors cite references in the experimental section.
4. why authors choose C57BL/6J mice as animal mode. It is better brifly introduce the propose in the manuscript.
5. I suggest rearrange fig 1.
6.Please add HE staining or other morpholgy assay.
7. Too much paragrphs in disscussion section, please revise them.
Author Response
Thank you for the thoughtful and constructive feedback you provided regarding our manuscript, Piceatannol prevents obesity and fat accumulation caused by estrogen deficiency in female mice by promoting lipolysis.
We concur with your observations and have integrated your suggestions into our manuscript.
Point 1. There are two 2.4 section, please revised it.
We have reflected this comment.
Point 2. No data statistics in the manuscript. How many reduplicate assays you did?
We replicated the Western blotting and qPCR experiments at least three times and validated the consistency of the outcomes. We have included a statement to this effect in the description of Figure 4 (Lines 266-267). And we have described the statistical analysis methods in the “Materials and Methods” section (Lines 169-173).
Point 3. I suggest the authors cite references in the experimental section.
We have supplemented the experimental section (Materials and Methods section) with cite references of previous report and brief explanatory notes.
- 2. Animals and experimental design (Lines 82-88)
- 5. Micro X-ray computed tomography (CT) analysis of fat accumulation (Lines 125-132)
Point 4. why authors choose C57BL/6J mice as animal mode. It is better briefly introduce the propose in the manuscript.
Thank you for your valuable comments. In the "Introduction" section, we have incorporated a previous study that highlights the distinct susceptibility of male and female C57BL/6J mice to dietary obesity. Moreover, we have emphasized, by citing relevant references, that C57BL/6J OVX mice are a well-suited model for estrogen-deficient obesity (Lines 51-60).
Point 5. I suggest rearrange fig 1.
Are you referring to the placement of Figure 1c specifically? We have resized and repositioned the three figures, which illustrate tissue weights, to fit within one column.
Point 6. Please add HE staining or other morphology assay.
Actually, we have previously tried HE staining in an experiment to see the dose dependent effect of PIC on OVX mice, and found that the difference was not very pronounced. In this study, tissue samples were preferentially used for the protein expression levels, so unfortunately, there were not enough samples left for HE staining.
Point 7. Too much paragrphs in disscussion section, please revise them.
We have reflected this comment.
Thank you once again for your consideration of our paper.

Reviewer 2 Report
The manuscript "Piceatannol prevents obesity and fat accumulation caused by estrogen deficiency in female mice by promoting lipolysis" is very interesting and investigates the effect of Piceatannol (PIC) on postmenopausal obesity and its mechanism of action using a mouse model. The authors successfully highlight the importance of postmenopausal obesity and the potential of PIC as an anti-obesity agent. The findings are relevant to the field of obesity research and could have important implications for the development of new therapies to treat postmenopausal obesity.
Comments for authors:
1. The authors should discuss the role of PGC-1α, an important gene that regulates lipid metabolism and mitochondrial biogenesis, in the discussion section to help promote the understanding of PIC's function in lipid regulation in this model.
2. Please reorganize the sentence in line 42.
3. In line 153, "High fat" should be in lowercase.
4. In line 155, "Adipose" should be in lowercase.
5. Figure 4(d) is missing a bar.
6. Please be consistent with the use of "Western blot" or "Western blotting" throughout the manuscript.
Overall, I recommend this manuscript for publication after the minor revisions are made. The study's methodology is robust, and the results are well-supported by the data presented. Although there are minor errors that need to be addressed, the manuscript is written well and provides important insights into the potential of PIC as an agent to inhibit fat accumulation caused by menopause.
Author Response
Thank you for the thoughtful and constructive feedback you provided regarding our manuscript, Piceatannol prevents obesity and fat accumulation caused by estrogen deficiency in female mice by promoting lipolysis.
We concur with your observations and have integrated your suggestions into our manuscript.
Point 1. The authors should discuss the role of PGC-1α, an important gene that regulates lipid metabolism and mitochondrial biogenesis, in the discussion section to help promote the understanding of PIC's function in lipid regulation in this model.
As you pointed out, we have incorporated a statement in the Discussion section, noting the need for further research on the activation of PGC-1α and the anti-obesity effects of PICs, given their potential relationship with energy metabolism and mitochondrial production (Lines 348-355).
Point 2. Please reorganize the sentence in line 42.
We have restructured the text in the following manner (Lines 42-43):
Previous studies have identified many biological activities of PICs, including antioxidant [19], anticancer [20], and anti-inflammatory effects [11,21,22].
Point 3. In line 153, "High fat" should be in lowercase.
Thank you for pointing this out. The revised manuscript now uses the abbreviation "HFD" when referring to the "high-fat diet" (Line 208).
Point 4. In line 155, "Adipose" should be in lowercase.
We have reflected this comment (Line 210).
Point 5. Figure 4(d) is missing a bar.
We have reflected this comment.
Point 6. Please be consistent with the use of "Western blot" or "Western blotting" throughout the manuscript.
We have reflected this comment (The notation has been unified to "Western blotting").
Thank you once again for your consideration of our paper.
